# Feasibility of Using the Epidural Space Detecting Device (EPI-Detection^TM^) for Interlaminar Cervical Epidural Injection

**DOI:** 10.3390/jcm9082355

**Published:** 2020-07-23

**Authors:** Jiin Kang, Sam Sun Park, Chul Hwan Kim, Eui Chul Kim, Hyung Cheol Kim, Hyungseok Jeon, Kyung Hyun Kim, Dong Ah Shin

**Affiliations:** 1Department of Neurosurgery, Yonsei University College of Medicine, 50 Yonsei-ro, Seodaemun-gu, Seoul 03722, Korea; jiinhanes@yuhs.ac (J.K.); SSAMI332@yuhs.ac (S.S.P.); HANGABB@yuhs.ac (C.H.K.); YEOI81@yuhs.ac (E.C.K.); KIMHC@yuhs.ac (H.C.K.); SISUNSINGUM@yuhs.ac (H.J.); 2Department of Neurosurgery, Yongin Severance Hospital, Yonsei University College of Medicine, 363, Dongbaekjukjeon-daero, Giheung-gu, Yongin-si, Gyeonggi-do 16995, Korea; 3Department of Neurosurgery, Gangnam Severance Hospital, Spine and Spinal Cord Institute, Yonsei University College of Medicine, 211, Eonju-ro, Gangnam-gu, Seoul 06273, Korea

**Keywords:** epidural injections, cervical radiculopathy, epidural space, epidural anesthesia, complications

## Abstract

Cervical epidural injection (CEI), which is widely used for the treatment of cervical radiculopathy, sometimes has been associated with post-operative complications. Recently, EPI-Detection^TM^, which detects the negative pressure of the epidural space and notifies the proceduralist by flashing a light and producing a beeping sound, was introduced. We assumed that the newly developed device could be as safe and efficient as the conventional loss of resistance (LOR) method. Therefore, we aimed to evaluate the effectiveness of the EPI-Detection^TM^ and compare it to that of the conventional LOR method. We randomly assigned 57 patients to the LOR and EPI-Detection groups (29 and 28 patients, respectively). Subjects were treated with interlaminar CEI (ILCEI) using one of two methods. The measured parameters, i.e., operation time and radiation dose were lower in the EPI-Detection^TM^ group (4.6 ± 1.2 min vs. 6.9 ± 2.1 min; and 223.2 ± 206.7 mGy·cm^2^ vs. 380.3 ± 340.9 mGy·cm^2^, respectively; all *p* < 0.05) than in the LOR group. There were no complications noted in either group. Both the EPI-Detection^TM^ and LOR methods were safe and effective in detecting the epidural space, but the former was superior to the latter in terms of operation time and radiation exposure. The EPI-Detection^TM^ may help perform ILCEI safely.

## 1. Introduction

Epidural injection is the most commonly used interventional technique for the management of spinal pain [1]. The use of epidural injections has increased by 130%, with an annual increase of 7.5% in the U.S in 2000s [1]. This increasing trend applies to cervical epidural injection (CEI) as well. Many patients are subjected to CEI because of an increasingly aging population [1]. The efficacy of cervical epidural injection (CEI) has been demonstrated by previous studies [2,3,4,5,6,7,8]. Manchikanti et al. reported that CEI significantly reduced pain and disability in patients with cervical disc degeneration, cervical stenosis, and post spinal surgery syndrome [4,7,8]. Stav et al. reported that CEI was superior to muscular injection in the management of neck pain in terms of better pain relief and work recovery [2].

CEI includes transforaminal (TF) and interlaminar (IL) approaches. In TFCEI, a spinal needle is advanced near the dorsal root ganglion under fluoroscopic guidance, and the tip position is finally confirmed by the spread of contrast agent. In ILCEI, loss of resistance (LOR) is used between fluoroscopy-guided advancement and confirmation of contrast agent spread. Although radicular artery puncture can be avoided by using ILCEI, this method still has potential complications. Abbasi et al. reported that the overall incidence of ILCEI-related complications ranges from 0–16.8% [9]. Improper insertion of the spinal needle through the interlaminar space can lead to venous plexus perforation, intrathecal injection, and intramedullary injection. These may lead to irreversible neurological dysfunction and life-threatening conditions [9,10,11,12,13,14]. Therefore, it is urgent to develop an accurate and safe method for identifying the location of epidural space during ILCEI procedure.

To date, LOR is the most commonly used method to detect the epidural space [15]. In the LOR technique, a physician feels the disappearance of resistance with his fingertip when a spinal needle penetrates the ligamentum flavum. However, the ligamentum flavum may be missing in the middle of the interlaminar space, and there is a difference in the sensitivity of LOR depending on the physician’s ability [16,17,18]. Thus, procedural errors can potentially occur. Recently, devices for locating the epidural space on the basis of its intrinsic negative pressure have been introduced. The Epidrum^TM^ (Exmoor Innovations Ltd., Taunton, UK) has a transparent balloon. The balloon is filled with 1–1.5 mL of air just before ILCEI. When a spinal needle connected with the Epidrum^TM^ is advanced into the epidural space, the inflated balloon collapses confirming the correct placement of the epidural catheter [19,20,21,22]. The Epi-Jet^TM^ (Egemen International, Turkey) is a syringe device that has a piston filled with 4–6 mL of air or fluid. When a spinal needle connected with the Epi-Jet^TM^ reaches the epidural space, the piston unlocks and deflates allowing a physician to confirm the epidural space [19].

While both the Epidrum^TM^ and Epi-Jet^TM^ administer a small amount of air or normal saline into the epidural space, the EPI-Detection^TM^ (SAEUM Meditec Co., Ltd., Bucheon, South Korea) detects intrinsic negative pressure by a specialized piezo-resistive micromachined senor. When it locates the epidural space, it informs the physician by producing a beeping sound and by flashing light. We assumed that the EPI-Detection^TM^ device could be as safe and efficient as the conventional LOR method in ILCEI. The purpose of this study was to evaluate the safety and efficacy of the EPI-Detection^TM^ in comparison to that of the LOR method in ILCEI.

## 2. Materials and Methods

### 2.1. Study Design

This was a single-center, randomized, single-blinded study performed according to the guidelines for randomized controlled trials. The protocol was approved by the institutional review board (1-2017-0077). The study was also conducted in accordance with the tenets of the Declaration of Helsinki. The design consisted of a control and experimental group. The LOR group (control group) was treated with ILCEI using the LOR technique. The EPI-Detection^TM^ group (experimental group) was treated with ILCEI using the EPI-Detection^TM^ technique. Primary endpoints were procedure time, radiation dose, and failure rate. Secondary endpoints were visual analog scale (VAS, 0–10) pain score, MacNab scale, and occurrence of complications. With an effect size of 10%, significance level of 5%, and power of 80%, the sample size was calculated. When the difference in success rate between the two groups was expected to be within 25%, and the patient drop rate was estimated to be maximum 25%, 35 sample sizes were needed per arm.

### 2.2. Patient Selection and Blinding Strategy

Seventy consecutive patients were enrolled from January 2018 to January 2019 (Figure 1). Inclusion criteria were chronic posterior neck or arm pain in patients aged from 18 to 70 years. All patients experienced significant pain and a VAS score of 5 or more after receiving appropriate conservative treatment for at least 6 months in the form of medication and/or physiotherapy. Patients with myelopathy, neurological dysfunction, spondylitis, tumor, spine fracture, elevated WBC count, and coagulopathy, and those who were pregnant or lactating were excluded. Thirteen patients were excluded on the basis of the exclusion criteria and because they declined to participate. Finally, a total of 57 patients were randomly assigned to one of the two groups: the LOR (29 cases) or EPI-Detection^TM^ (28 cases) group.

A randomization schedule with 1:1 ratio was generated using a computer program (SAS 9.3, SAS Institute Inc., NC, USA). After the patient entered the operating room, the group to which the patient belonged was notified to the operator. One single physician (D.A.S.) with more than 20 years of experience performed CEI on the patient. The operator, however, treated both groups strictly and fairly.

Since this study is a randomized single-blinded study, a blinding strategy is essential. To reduce the bias of the operator according to the patient’s physique and gender, the operator was able to know whether to use the LOR method or the EPI-Detection ^TM^ method just before the procedure. In addition, in order to reduce the bias of the endpoint according to the radiation dose a, the radiology technician was not informed about the experiment. Even if the radiology technician did not know the outcome, the radiation doses were automatically recorded in fluoroscopy so that the data manager could obtain information later. Also to prevent the bias of the patient’s clinical outcome, the patient was not informed whether the EPI-Detection^TM^ was used or the LOR method was used. After each procedure, all data were securely stored in a designated data base. Only independent evaluators who have not participated in the patient’s injection process can access the data.

### 2.3. EPI-Detection^TM^ Device

The EPI-Detection^TM^ is a device that detects the negative pressure in the epidural space. When a needle tip reaches the epidural space, it beeps and flashes a light. The system consists of a connection pipe connected to a spinal needle, a piezo-resistive pressure sensor, a small circuit board processing digital signals, a speaker that produces a beep sound, an LED that emits light of different colors depending on the condition, and a luer lock connected to a therapeutic syringe or catheter (Figure 2). The allowable error of the device ranges from −0.5 mbar to +0.5 mbar. The pressure sensor detects the minute negative pressure that occurs when the needle penetrates the ligamentum flavum and opens the epidural space. This process does not involve the injection of any material; it only detects pressure gradient in the air that occurs naturally. The optimal pressure was determined in our previous study [23]. While advancing the needle, the cap is locked to secure the epidural space and the device is closed. After the needle tip is located in the epidural space, the locked cap can be released to inject the drug or to advance a catheter.

### 2.4. Interlaminar Epidural Injection

All procedures were performed aseptically while monitoring each patient in the operating room. Each patient was placed in the prone position on a radiolucent table, with a soft pillow on his/her chest, his/her forehead on the table, and his/her neck bent slightly to open the interlaminar space. All patients underwent ILCEI at the level of C7-T1, where the epidural space is the largest [24,25]. After disinfection of the skin with betadine, the skin was anesthetized with 1 mL of 1% lidocaine at the entry point. A 20-gauge Tuohy needle was slowly inserted into the midline under fluoroscopy guidance. After reaching the spinolaminar line, the predetermined LOR or EPI-Detection^TM^ procedure as described separately in the sections below was performed to place the tip of the Tuohy cannula into the epidural space.

### 2.5. EPI-Detection^TM^ Procedure

The plastic stylet was removed, leaving only the Tuohy cannula. The connection pipe of the EPI-Detection^TM^ device was connected to the hub of the Tuohy cannula. The operator checked whether the luer-lock part was securely locked with the cap. The Tuohy cannula connected with the EPI-Detection^TM^ was slowly advanced into the epidural space under fluoroscopic guidance. When the device sensed negative pressure in the epidural space, the light changed from green to blue and produced a beeping sound (Figure 3). Subsequently, 1 mL of contrast dye (Omnipaque 300, GE Healthcare, IL, USA) was injected through the needle under fluoroscopy to confirm the epidural space.

### 2.6. LOR Procedure

A 5-cc syringe filled with 2-cc of air was connected to the hub of the Tuohy cannula. While slowly advancing the cannula, the operator felt the pressure while gently pressing the plunger with his thumb. The needle was slowly and steadily advanced. As soon as the pressure suddenly disappeared, the advancement of the needle was stopped. As in the case of the EPI-Detection^TM^ group, 1 mL of contrast dye was injected. Fluoroscopy was also performed to confirm the epidural spread of the dye.

### 2.7. Outcomes Assessment

After confirming that the dye had spread to the epidural space and did not extravasate, 0.2% ropivacaine (5 cc) and dexamethasone 5 mg (1 cc) were slowly injected. The patient was observed for 30 min in the recovery room and discharged if there were no specific problems. Outcomes were assessed postoperatively at 1 day using the VAS score of the neck or arm pain and at 1 month using the MacNab criteria for the functional outcomes. The MacNab scale was devised by Orthopedic surgeon Ian MacNab, and is layered into 4 levels according to the patient’s well-being after surgery and procedure. “Excellent” is defined as “No pain; no restriction of activity”. “Good” is defined as “Occasional pain of sufficient severity to interfere with the patient’s ability to do his normal work or his capacity to enjoy himself in leisure hours”. “Fair” is defined as “Improved functional capacity, but handicapped by intermittent pain of sufficient severity to curtail or modify work or leisure activities”. “Poor” is defined as “No improvement or insufficient improvement to enable increase in activities; further operative intervention required”. Operation time (min) was measured from the completion of aseptic draping to the removal of the needle after epidural injection. Radiation dose (mGy·cm^2^) during the procedure was measured by C-arm fluoroscopy (GE OEC 9800 Plus, GE OEC Medical system, GE Healthcare, IL, USA). After fluoroscopy, radiation information of dose and time for each patient were recorded in the storage medium of OEC 9800 plus for each time period. In “Dose summary mode” of GE OEC 9800 Plus, we can obtain information about the patient’s radiation dose and time. We also investigated procedure-related complications like motor deficit, mono-, para- and quadriplegia.

### 2.8. Statistical Analysis

Chi-square test and independent t-test were used to compare demographic data of study participants and outcomes. Linear regression analyses were performed to analyze operation time and radiation dose. All statistical analyses were performed using SPSS (SPSS Inc., Chicago, IL, USA), and statistical significance was defined as *p* < 0.05.

## 3. Results

Among the 70 participants screened, 57 patients (39 men and 18 women) were included in the final experiment. Among them, 29 were assigned to the LOR group and 28 to the EPI-Detection^TM^ group. The subjects’ age ranged from 30–77 years with a median of 56 years. There were no statistically significant differences in sex, age, height, weight, BMI, and diagnosis between the two groups. The diagnoses of the patients were as follows: clearly diagnosed herniated cervical disc and unidentified chronic neck pain that was unresponsive to various treatments. The affected levels ranged from C4–5 to C6–7. The epidural injections, however, were administered to the C7-T1 level regardless of the affected levels in all patients because the epidural space is the widest in that level, so that injectates can spread to the target level. Demographics and baseline clinical data are shown in Table 1. All patients were discharged within 24 h of undergoing the assigned procedure.

The operation time and radiation dose were 35% and 42% lower, respectively, in the EPI-Detection^TM^ group than in the LOR group (4.6 ± 1.2 min vs. 6.9 ± 2.1 min, *p* = 0.000; 7.5 ± 3.4 s vs. 11.4 ± 6.5 s, *p* = 0.007; 223.2 ± 206.7 mGy·cm^2^ vs. 380.3 ± 340.9 mGy·cm^2^; *p* = 0.040; Table 2). All the treatment outcomes were “good” or “excellent” according to the MacNab criteria, indicating that they were successful. No complications were observed during or after the procedure in either of the groups (Table 2).

To compare the operation time, regression analysis was performed based on the procedure type and demographic data. Among these variables, only the use of the EPI-Detection^TM^ method was predictive of a shorter operation time (R^2^ = 33.5, *p* < 0.05; Table 3).

## 4. Discussion

In this study, 59 patients were randomly selected, and the LOR method and EPI-Detection^TM^ method were applied to perform ILCEI. There were no statistical differences in the demographics and the baseline clinical data between the LOR group and the EPI-Detection^TM^ group. However, in the group using EPI-Detection^TM^, operation time and dose were statistically significantly decreased compared to LOR group. However, there was no statistical difference between prognosis and complications.

Neck pain is a common condition showing a lifetime prevalence ranging from 14.2% to 71%, with a mean of 48.5% [26]. Considering their favorable outcomes, less invasive treatments are initially preferred. If the pain is not alleviated by conservative treatment consisting of life-style modification, analgesic use, and physical therapy, epidural administration of anesthetics with or without steroids may be used. The success rate of CEI has been reported to range from 14% to 92%, depending on the diagnosis, technique used, and outcome measure considered [24,27,28,29,30,31,32,33,34]; Slipman et al. reported a low success rate of 14–20% when CEI was administered to manage trauma-related cervicobrachialgia. However, this indication is rarely used currently, and the follow-up period in Slipman’s study was more than 20 months after CEI [35,36]. Randomized trials on CEI have shown favorable outcomes [2,37]. CEI has been widely accepted and its use has increased rapidly. In this study, a total of 57 consecutive patients were included. Among them, 67% reported excellent results one month after the procedure. This result is comparable to that of previous studies.

Epidural injection is highly effective in the management of spinal pain, but not without complication. In particular, cervical procedures can cause fatal complications. These complications include simple postoperative pain, vasovagal syncope, cerebrospinal fluid leakage, infection, epidural hematoma, pneumocephalus, spinal cord infarction, and direct spinal cord injury [9,10,11,12,13,14,38,39,40,41,42,43,44,45,46,47]. Most of them are caused by needle misplacement. Because of the severity of these complications, a physician must remain attentive at all times during the procedures, which have steep learning curves. Conventionally, the epidural space has been detected using the LOR method, assuming that there is a difference in friction when the needle penetrates the ligamentum flavum and that the epidural pressure is negative. However, there is a potential danger because the ligamentum flavum is sometimes missing in the midline, and the procedure depends on a physician’s subjective sensations. To overcome these drawbacks, various probing devices and training simulators have been developed [19,20,21,22,48,49,50,51]. In line with this idea, the EPI-Detection^TM^ was developed. Our study showed that CEI was more likely to be successful in the first attempt when the EPI-Detection^TM^ method was used than when the LOR method was used (96% vs. 86%). However, this study did not prove that the LOR method was associated with a higher incidence rate of complications than was the EPI-Detection^TM^ method. This is probably because the sample size is small and CEI was performed by an expert, not a beginner.

The existence of negative pressure in the epidural space has been reported since Jansen’s first report. Zarzur et al. studied 30 patients who underwent epidural anesthesia and found that the pressure within their epidural space was 0 to −0.16 psi [52]. Lee et al. reported that the optimal pressure value that helps detect the epidural space was −5 mbar [23]. The EPI-Detection^TM^ detects the negative pressure in the epidural space and instantly notifies a physician by flashing a light and producing beeping sound. In this study, the use of Epi-Detection^TM^ reduced the operation time by 34%, relative to the control group. In the LOR group, we assumed that the operation time was longer because of frequent checking of the position of the needle tip by alternating between AP and lateral views. This assumption could be made because the radiation time in the LOR group was longer. Our linear regression analysis revealed that the use of the EPI-Detection^TM^ was the only variable significantly associated with operation time. There are studies whose findings a similar to those observed in our study. Mittal et al. reported that the acoustic puncture assist device, which is equipped with a pressure meter and displays real-time pressure when connected to the syringe, significantly reduces operation time, relative to the LOR method (19 s vs. 48 s) [53,54]. Kartal et al. showed that the Epidrum^TM^ method allowed for a quick and accurate entry into the epidural space [19].

Nonetheless, we do not believe that EPI-Detection^TM^ method will always ensure procedure success or that it can completely replace fluoroscopy. Unless epidural injection is performed carefully regardless of the use of any aiding device, serious injury may still occur. Some reports argued that no negative pressure exists in the epidural space and that this negative pressure is temporarily observed owing to momentary displacement of the ligamentum flavum [52,55]. This negative pressure would not exist if the epidural space was severely damaged by surgery, infection, or mass. Kartal et al. showed that the Epidrum^TM^ method had a relatively high failure rate [19]. Other studies have also reported that the device is unreliable, depending on the location and the number of punctures [51]. However, the research on EPI-Detection^TM^ use is in the early stage; therefore, judging its usefulness should be withheld until a large-scale study is conducted. Therefore, we insist that it should be used with care in its current state. In other words, it is only a supplement, and the use of fluoroscopy and caution are still absolutely necessary.

The present study has several limitations. First, the sample size was small. Although the sample size was statistically calculated, the number of participants was small, because of which we could not reveal any significant differences in complication rates between the groups. There is also a result that the sample size is too small for the results of this study to discuss the normality of the distribution. Further research is needed, including more sample sizes. Additionally, this study is a single blind study. In a single blind study, the bias of the researcher can influence the outcome. This is the biggest limitation of this study. Third, only cervical levels were included in this study. The lumbar level at which epidural injections are most frequently performed should be included in future studies. Finally, only one physician performed CEI in this study. The complication rate could differ depending on the physician performing the procedure and his/her experience level. In the future, large-scale multi-center studies should be conducted to address these problems. Despite these limitations, our findings suggest that the less operation time associated with EPI-Detection^TM^ use will reduce procedural burden on physicians and patients and that reduced radiation exposure will contribute to safety of the procedure.

## 5. Conclusions

Both the EPI-Detection^TM^ and LOR methods were safe and effective for detecting the epidural space during ILCEI. The Epi-Detection^TM^ method was superior to the LOR method in terms of operation time and radiation exposure. The EPI-Detection^TM^ method may help perform ILCEI safely.

## Figures and Tables

**Figure 1 jcm-09-02355-f001:**
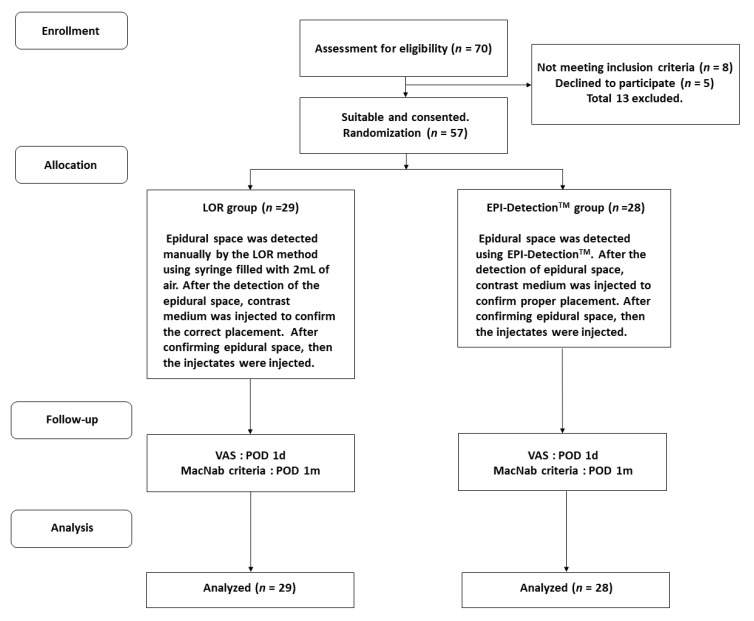
Flow diagram showing the method of patient recruitment. LOR, loss of resistance; VAS, visual analog scale; POD, post-operative day.

**Figure 2 jcm-09-02355-f002:**
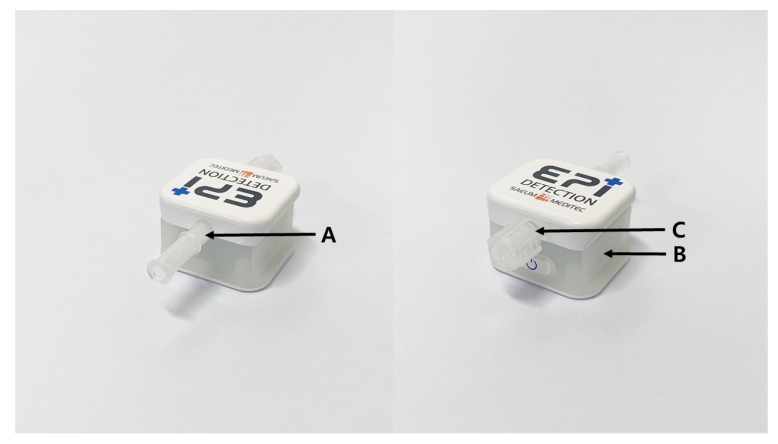
EPI-Detection^TM^ device. It consists of a connection pipe (**A**) connected to a spinal needle, a piezo-resistive pressure sensor, a small circuit board processing digital signals, a speaker that produces beep sound, an LED (**B**) that emits different light depending on the condition, and a luer lock (**C**) connected to a therapeutic syringe or catheter connected to a Tuohy needle.

**Figure 3 jcm-09-02355-f003:**
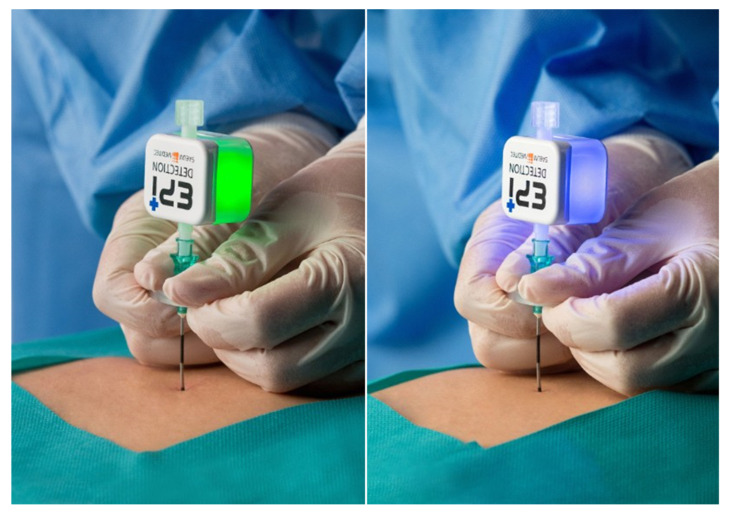
In the ready-on state, the LED glows green. As the needle advances and reaches the epidural space, the LED turns blue and produces a beep sound.

**Table 1 jcm-09-02355-t001:** Summary of patient demographic and baseline clinical data.

	LOR (*n* = 29)	EPI-Detection^TM^ (*n* = 28)	*p*-Value
Sex (M:F)	22:7	17:11	0.219
Age (years)	54.2 ± 9.5	53.6 ± 11.2	0.848
Height (cm)	167.2 ± 8.7	166.6 ± 9.6	0.887
Weight (kg)	68.0 ± 11.5	67.6 ± 11.1	0.784
BMI	24.2 ± 2.8	24.2 ± 2.1	0.967
BMI > 25 (yes: no)	9:20	13:25	0.233
Preoperative diagnosis			
Herniated cervical disc (%)	65.5% (19)	67.9% (19)	0.851
Chronic neck pain (%)	34.5% (10)	32.1% (9)	
Treated level			
C4–5 (%)	41.4% (12)	32.1% (9)	0.609
C5–6 (%)	34.5% (10)	32.1% (9)	
C6–7 (%)	24.1% (7)	35.7% (10)	

LOR, loss of resistance; BMI, body mass index. Data were compared by Chi square tests and independent t-tests.

**Table 2 jcm-09-02355-t002:** Comparison of the operation time and radiation dose.

	LOR (*n* = 29)	EPI-Detection^TM^ (*n* = 28)	*p*-Value
Operation Profile			
Operation time (min)	6.9 ± 2.1	4.6 ± 1.2	0.000 *
Radiation dose (mGy·cm^2^)	380.3 ± 340.9	223.2 ± 206.7	0.040 *
Failure of epidural puncture on the first try (%)	4 (13.8)	1 (3.6)	0.173
Outcome Scale			
VAS score on POD 1	2.4 ± 1.3	1.9 ± 1.8	0.153
MacNab Scale score			
Excellent	18	20	0.454
Good	11	8	
Complications (%)	0 (0)	0 (0)	1.000

LOR, loss of resistance; VAS, visual analog scale; POD, postoperative day. Data were compared using Chi square test and independent t-test. * *p* < 0.05.

**Table 3 jcm-09-02355-t003:** Factors predictive of total operation time with regard to cervical epidural steroid injection.

	B	ß	95% CI	*p*-Value
Age	−0.570	−0.046	−4.012, 2.873	0.741
Sex (female)	39.159	0.145	−206.029, −88.784	0.399
Weight	0.033	0.003	−4.502, 4.569	0.988
Height	0.807	0.058	−5.629, 7.242	0.802
EPI-Detection^TM^ (Yes)	−147.406	−0.589	−206.019, −88.784	0.000 *

CI, confidence interval. Data were analyzed by linear regression, * *p* < 0.05, R^2^ = 33.5.

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
