# Peer review of "Feasibility of Using the Epidural Space Detecting Device (EPI-DetectionTM) for Interlaminar Cervical Epidural Injection"

_jcm, 2020, doi:10.3390/jcm9082355_

Round 1
Reviewer 1 Report
Dear authors,
Thanks for submitting your work to the journal.
Globally, your work is well done, relevant and clearly presented.
The statistics should be checked, especially the tests assumptions (normality of the distributions).
Please avoid stating about differences when these are not statistically significant. This is essential. The maximum that you might to is discuss potential differences not identified because of underpowering.
Reviewer 2 Report
Feasibility of Using the Epidural Space Detecting Device (EPI-Detection™) for Interlaminar Cervical Epidural Injection
Overall, a well-performed randomized clinical study, and I believe it is important to consider publications. I only have several minor critiques.
L22: “surgeon” – change to “proceduralist” as there are non-surgery physicians who will perform the procedure, including anesthesiologists.
L82, L109: “case-control” – delete.
L81: Study design – the authors listed three outcomes as the primary outcomes. Then calculated 10% effect size ad other conditions to yield 35 cases per group. First, the radiation dose is the product of radiation time, therefore, it is unnecessary to indicate the radiation time (or radiation time) as the primary outcomes. Second, using the two outcomes as the primary outcomes is usually tricky to calculate the sample size. Please clarify how to calculate the sample size using the two indices (assuming the first point above is accurate).
L82: Single-blind manner is the only way to perform this study; however, to be strict, the proceduralists should also blind to the intention of the study as there may be bias for the procedure outcomes. Please elaborate on the study condition in this regard.
L106: “One identical physician” – What do you mean? Did a single surgeon perform all procedures? Please identify the surgeon using the initial, if he/she is among the authors.
L102: ”Finally, a total of 57 patients were randomly assigned to one of the two groups:” - normally, you would like to have at least 35 patients each group to fulfill the sample calculations to avoid a beta error. Why didn’t the authors approach more patients? It is just lucky to have all outcomes turned out to reach a statistical significance.
L113: “the radiologist was not informed about the experiment” – I am confused. How was a radiologist (physician) involved in the study? Was he/she attending the procedure? Usually, there is no need for additional personnel other than the proceduralist as an MD. You may need a radiology technician. Please clarify how you got the data of the radiation time and exposure dosage, too.
L188: “independent t-test” – what does it mean? How did you compare the primary outcomes?
L189: “Linear regression analyses were performed to analyze operation time and radiation exposure.” – Only operation time’s regression analysis was presented in the result section. How did you determine the factors that influence each outcome?
L210: “The LOR group showed a higher failure rate of epidural puncture on the first try” – define this outcome in the method section.
Table 2: “MacNab Scale” – define this scale in the method section.
Author Response
Please See the attachment.
Thank you very much.

This manuscript is a resubmission of an earlier submission. The following is a list of the peer review reports and author responses from that submission.
Round 1
Reviewer 1 Report
Scientific soundness was rated as averagebecause of a lack of description of the blinding procedures. The paper needs a few details about the blinding of the physician. Clearly the physician knew which procedure that was performed. However did the physician know that the main outcome measure was time required to complete the procedure? If this was known, it is possible that the time required to complete the procedure was influenced by the physicians expectations or preference of procedure. Furthermore was the radiologist aware that outcomes included radiation time and radiation dose? The time would likely be dictated by the physician performing the procedure, however the radiologist would be controlling the kVP and mA and hence radiation dose.
If the blinding is described in greater detail and it turns out that neither the physician or the person responsible for imaging were in fact blinded to the outcomes, then the scientific soundness improves.
Other than blinding, there are just a few edits required as follows:
Line 20 The abbreviation LOR is used for the first time without defining it. It is not defined until the introduction
Line 26 "There was no complication in both groups," should be replaced with "There were no complications in either group."
Line 40 I'm not sure what you mean by "Due to the aging of patients...has increased." I suspect you are indicating that the baby boomer cohort has reached an age in which cervical spine pathology has been more frequently observed in the past few years. So it should probably say "Due to the fact that a greater proportion of the population is older....has increased."
Line 113 "...one same physician..." should read "...the same physician..."
Line 145 Should read "There were no statistically significant demographic differences..."
Line 152 should read "...this difference was statistically significant..."
Line 161 3.6 is not more than 4 times greater than 13.8 as you indicate. It is slightly less that 4 times greater.
Line 168 R2 should read R2
Line 188. "Some researches have..." should read "Some research has..."
Line 222 "Due to the accuracy of the EPC and easy noticeability,
in more than 96% of cases using EPC, EDS could be detected at one time in our study. " should read "Due to the accuracy of the EPC and ease of application, the EDS was detected on the first attempt in 96% of the cases utilizing the EPC procedure as compared to 86% in the LOR procedure. However this difference was not statistically significant."
Line 229 "In addition, the study that confirms the difference and
reliability between physicians is needed through a large-scale study with increased physicians." should be changed to "In addition studies should be done to assess the reliability of the EPC procedure between physicians."
Line 232 "If studies with a large number of patients or with a larger number of physicians are conducted, the rate of complications will be different with this study." should be changed to "In order to assess the rate of complications for the EPC compared to LOR, studies should be done on a larger number of patients utilizing a greater number of physicians."
Reviewer 2 Report
The title of the paper is described as 'Utility of an Epidural Pressure Checker for Cervical 2 Epidural Steroid Injection'. It is actually an RCT comparing 2 different techniques to identify cervical epidural space. The manuscript and abstract are not structured or reported in a way that is appropriate to the method. The rationale for doing this is unclear. The background suggests that CESI is important to manage cervical radicular pain. It also describes that complications can happen as a result of CESI, but most of the complications that are reported does not happen as a result of not being able to identify the space. How can you support your statement that better identification will decrease those complications? Epidural hematoma happens because of bleeding, which can still happen with the device to identify EDS. Similarly, paraplegia and other neurological complications shown in literature are largely multifactorial, potentially caused by particulate steroid, which is again not prevented by EDS device. The authors never describe their study objectives. Even if you consider complications can be prevented or minimized, your outcomes are not reflective of comparing those complications but are just related to technique-such as procedural time. Literature is very suggestive that identification of CES should be performed using fluoroscopy and in fact identification of CES by relying on other methods could be dangerous as the ligamentum flavum is thinnest at the C7-T1 cervical level, it being usually around 2–3 mm, and may not be completely fused in a majority of patients. This was demonstrated in a study on 50 human cadavers and found that in C7-T1 alone, the incidence of midline gaps was 51%, which increased to 64% at C6-C7 [Lirk P, Kolbitsch C, Putz G, Colvin J, Colvin HP, Lorenz I, Keller C, Kirchmair L, Rieder J, Moriggl B. Cervical and high thoracic ligamentum flavum frequently fails to fuse in the midline. Anesthesiology. 2003 Dec;99(6):1387-90] and [Stojanovic MP, Vu TN, Caneris O, Slezak J, Cohen SP, Sang CN. The role of fluoroscopy in cervical epidural steroid injections: an analysis of contrast dispersal patterns. Spine (Phila Pa 1976). 2002 Mar 1;27(5):509-14]. The authors make no attempt to estimate their sample size necessary. The patient selection criteria is not described appropriately. The methods to control bias [randomization, allocation concealment, blinding] is not described. In view of of all the above factors, I think it is very hard to convince practitioners to use the Epidural to avoid complications arising while performing CESI. There are clearly many devices that help assist identification of epidural space, both lumbar and cervical [Teng WN, Tsou MY, Chang WK, Ting CK. Eyes on the needle: Identification and confirmation of the epidural space. Asian J Anesthesiol. 2017 Jun;55(2):30-34]. Hence, it is not clear what this study would add more to the existing literature.